# Security and Privacy Analysis of Smartphone-Based Driver Monitoring Systems from the Developer’s Point of View

**DOI:** 10.3390/s22135063

**Published:** 2022-07-05

**Authors:** Dmitry Levshun, Andrey Chechulin, Igor Kotenko

**Affiliations:** St. Petersburg Federal Research Center of the Russian Academy of Sciences (SPC RAS), 199178 St. Petersburg, Russia; chechulin@comsec.spb.ru (A.C.); ivkote@comsec.spb.ru (I.K.)

**Keywords:** information security, intelligent transportation systems, security analysis, privacy analysis, white-box testing, driver monitoring systems, smartphone sensors

## Abstract

Nowadays, the whole driver monitoring system can be placed inside the vehicle driver’s smartphone, which introduces new security and privacy risks to the system. Because of the nature of the modern transportation systems, the consequences of the security issues in such systems can be crucial, leading to threat to human life and health. Moreover, despite the large number of security and privacy issues discovered in smartphone applications on a daily basis, there is no general approach for their automated analysis that can work in conditions that lack data and take into account specifics of the application area. Thus, this paper describes an original approach for a security and privacy analysis of driver monitoring systems based on smartphone sensors. This analysis uses white-box testing principles and aims to help developers evaluate and improve their products. The novelty of the proposed approach lies in combining various security and privacy analysis algorithms into a single automated approach for a specific area of application. Moreover, the suggested approach is modular and extensible, takes into account specific features of smartphone-based driver monitoring systems and works in conditions of lack or inaccessibility of data. The practical significance of the approach lies in the suggestions that are provided based on the conducted analysis. Those suggestions contain detected security and privacy issues and ways of their mitigation, together with limitations of the analysis due to the absence of data. It is assumed that such an approach would help developers take into account important aspects of security and privacy, thus reducing related issues in the developed products. An experimental evaluation of the approach is conducted on a car driver monitoring use case. In addition, the advantages and disadvantages of the proposed approach as well as future work directions are indicated.

## 1. Introduction

Driver monitoring systems have been known for more than 15 years and have become more complex and available every day [1]. Such systems monitor the driver behavior and the environment inside and outside the vehicle using numerous sensors and cameras. The data obtained are used to prevent dangerous situations on the road and warn the driver. The recent scientific trend in the development of such systems is to use smartphone sensors (rear and front camera, accelerometer, microphone, etc.) to monitor the driver, while software of the monitoring system is represented as a smartphone mobile application [2].

It means that the whole driver monitoring system can be placed inside the smartphone of the driver of the vehicle, which has its own advantages and disadvantages. The main advantage is the mobility of the solution—the monitoring system moves with the driver and can potentially be placed in any vehicle. Moreover, placing it on a smartphone as an application provides the system with access to constant updates, cloud computing and services. If the computing power of the current smartphone is not enough, then it can be replaced with a more modern model, transferring all system settings without loss.

The main disadvantage is in the limited amount of sensors that are typically used in smartphones, while the list of sensors cannot be changed or improved. It means that not every functionality of the typical driver monitoring system can be implemented based on the smartphone sensors [3]. Moreover, the use of smartphones as well as cloud services introduces new security and privacy risks to the system [4,5]. For example, according to the Check Point Mobile Security Report [6], 97% of organizations in 2020 faced mobile threats, 46% had at least one employee download a malicious application, while 40% of the world’s mobile devices are inherently vulnerable to cyberattacks. According to the Amazon Web Services Cloud Security Report [7], the main concern remains misconfiguration of the cloud platform (71%), exfiltration of sensitive data (59%) and insecure application programming interfaces (54%). It is important to note that in this paper, privacy risks are considered as risks of a user’s personal data leakage. In turn, the personal data are defined as information that identifies or can be used to identify the individual. It means that the analysis of the data flow from the smartphone sensors is not considered as a privacy issue if the corresponding data are processed and stored in accordance with the legal requirements.

Moreover, because of the nature of the modern transportation systems, the consequences of the security issues in driver monitoring systems can be crucial, leading to threat to human life and health [8]. As multiple reports are showing, the current state of security and privacy in clouds and mobile applications is far from acceptable and requires many new solutions, which are taking into account that mobile and cloud technologies are constantly evolving.

Thus, the scientific problem to be solved is the contradiction that despite the large number of security and privacy issues discovered in smartphone’s applications on a daily basis, there is no general approach for their automated analysis that can work in conditions of lack of data and take into account specifics of the application area. Therefore, this work is aimed at developing the original approach for security and privacy analysis of smartphone-based driver monitoring systems from the developer’s point of view. The main goal of this approach is to help developers detect security and privacy issues in their products as well as to suggest appropriate solutions to those issues. Note that while the workflow of the developed approach is universal for mobile applications of different operating systems, platforms and architectures, their specifics are considered in detail during the vulnerabilities and weaknesses detection. Moreover, modern mobile application’s development frameworks (for example, Swiftic [9], React Native [10], Flutter [11], etc.) are already helping developers to prevent security and privacy issues with their best practices, so the developed approach should be considered as an additional verification tool.

The contribution to the research field can be divided into the data models and the approach for security and privacy analysis of smartphone-based driver monitoring systems. Let us consider the novelty of each contribution in more detail.

Unlike existing solutions, the developed data models are aimed at the representation of smartphone-based driver monitoring systems from the developer’s point of view for their subsequent security and privacy analysis. That is why the models are storing only the information that can be provided by the developers and is relevant for the detection of security and privacy issues. In the developed models, the following data are supposed to be requested from the developers: functionality of the product, agreement between the users and the product owners, access rights on the user’s smartphone that are required for the correct work of the product, source code of the product, its logs and traffic as well as requirements under the law to work with user’s private data. In addition, the attacker models are used to make the analysis process more precise and do not consider all possible security threats.

The novelty of the approach lies in the combination of multiple analysis algorithms for the automated detection of the security and privacy issues and suggestion of the solutions to them. In the developed approach, the following algorithms are working with the input data: detection of the possible and already covered security issues, analysis of the actual state of work with a user’s private data, analysis related to these data law requirements as well as analysis of user’s permissions to work with private data. Based on the output of those algorithms—detected security and privacy issues, corresponding measures are suggested to the developers with the help of additional algorithms. In addition, the developed approach can work in conditions when part of the input data are not provided by the developers. Since the lack of the input data affects the various stages of the approach in different ways, this fact is taken into account when the quality of the analysis is evaluated by another algorithm.

The paper is organized as follows. Section 2 considers the state of the art in the area of security and privacy analysis of driver monitoring systems. In Section 3, the original data models are presented. Section 4 describes the new approach for security and privacy analysis of smartphone-based driver monitoring systems. In Section 5, an experimental evaluation of the developed approach on a car driver monitoring use case is presented. Section 6 considers the advantages and disadvantages of the presented approach. In Section 7, the main conclusions are described, and future work directions are indicated as well.

## 2. Related Work

The main research directions in the analysis of security and privacy issues in smartphone-based systems can be divided into analysis of their functionality [12,13,14], configuration [15,16,17,18], source code [19,20,21], logs [22,23,24] and traffic [25,26,27] as well as the analysis of the documents that are defining the work with the user’s private data [28,29,30]. Let us consider each direction in more detail.

The analysis of the functionality of smartphone-based systems includes the analysis of their hardware [31] and software elements [32], interfaces [33], data transfer protocols [34], data extraction, storage and transfer processes [35], as well as their relationship with the information security threats [36]. The solution of this problem is in many ways similar to the problems that are solved by such approaches as Security by Design [37,38,39,40,41] and Secure Development Lifecycle [42,43,44,45,46]. The key difference is that such solutions perform the analysis of the functionality of ready-made systems instead of their step-by-step design. However, the task of detecting potential attack vectors to which the object of analysis may be subjected, based on the components and protection elements used by it, as well as the environment of its operation, remain similar. Note that for such approaches, it is important to analyze not only the components that are associated with the smartphone application itself but all applications, systems and services that are involved in the smartphone-based driver monitoring system—backend applications, cloud storage and services, infotainment systems, etc.

The analysis of the configuration of smartphone-based systems includes the analysis of platforms and versions of their hardware components, firmware and operating systems, as well as software applications used by them, to extract their relationship with vulnerabilities [47,48,49,50,51]. The solution of this problem is in many ways similar to the problems solved by risk analysis and assessment approaches [52,53]. In those approaches, based on the system’s configuration, a set of CPEs (Common Platform Enumerations [54]) is retrieved. After that, based on data from open vulnerability databases, such as NVD (National Vulnerability Database [55]), it becomes possible to link those CPEs with information about vulnerabilities related to them (CVE, Common Vulnerabilities and Exposures [56]) and weaknesses (CWE, Common Weakness Enumeration [57]). Then, based on the information about vulnerabilities, it becomes possible to analyze their metrics (CVSS, Common Vulnerability Scoring System [58]) as well as to develop approaches to combining those metrics to form an integral assessment. The key difficulty is the incompleteness of open databases, the inaccuracy of the transition between the device description and the CPE Uniform Resource Identifier (URI) set, as well as the limited application scope of CVSS metrics.

The analysis of the source code of the smartphone’s applications includes an analysis of the code’s architecture [59] and logic of its operation [60] as well as methods for identifying buffer overflows [61], memory leaks [62] and code inserts [63]. In addition, any operation that requires the extraction, storage or transfer of the user’s sensitive data is checked for compliance with the privacy policy [64,65]. Another important task is to analyze the interaction of applications as well as the environment for their work [66]. Note that the main challenge for the efficient source code analysis is various approaches aimed at changing the source code representation, for example, obfuscation [67]. In addition, the source code can be written in various programming languages as well as presented in binary form, which may lead to the need of reverse engineering methods [68].

Smartphone systems’ logs and traffic are often used for the anomaly [69] and attacks detection [70], including the detection of complex multistep attacks [71]. However, for the security and privacy analysis of the smartphone-based driver monitoring systems, they are mostly used for the analysis of the work with the user’s sensitive data [72,73]. It is required to check if these data are anonymized and/or encrypted, where it is stored and to what destinations it is transferred.

The main documents that are defining the work process with the user’s private data in smartphone-based driver monitoring systems are the agreement between the user of the system and its owner as well as requirements under the law that are described in the user’s country government documents. As a rule, the legal requirements have a higher priority, which means that the user agreement cannot violate them and must comply with these legal requirements. The goal of the analysis of those documents is to define what is allowed regarding the user’s private data, while the process of the analysis can be automated [74,75,76]. After the legal possibilities are known, they can be checked for compliance with the current state of the work with the user’s private data.

At the same time, it is important to note that research in the field of smartphone-based driver monitoring systems is mainly aimed not at ensuring the security and privacy of the product but at expanding its functionality to provide additional safety on the road. The main directions are as follows:Driver behavior tracking that, for example, aimed at detection of the driver’s drowsiness [77] and distraction [78], unfastened belt [79], etc.;Road situation tracking that, for example, aims at detection of road accidents [80], technical works [81], specific weather conditions [82], etc.;Context situation interpretation that, for example, aims at detection of the dangerous noise levels [83] and vehicle maneuvers [84], etc.;Integration with vehicle infotainment systems for the data storage, transfer, analysis and interpretation [85].

The summary of the contribution of modern approaches for security and privacy analysis of the smartphone-based driver monitoring systems is presented in Table 1. It classifies approaches by the object of the analysis and describes their input and output data.

Summarizing the above analysis, the following disadvantages of the modern approaches for security and privacy analysis of the smartphone-based driver monitoring systems can be mentioned:Most of the solutions are focused on the analysis of one aspect of the security or privacy of such systems, while the task of combining them into a single automated approach has not been studied enough;Work of security and privacy analysis approaches in conditions of lack or inaccessibility of data has not been fully explored;Most of the approaches do not include the evaluation of the quality of the performed analysis into their reports;Most of the available solutions do not take into account specific features of the smartphone-based driver monitoring systems.

It means that a general approach for solving the issue of security and privacy analysis of the smartphone-based driver monitoring systems has not been formed yet. Therefore, an original approach that combines various analysis algorithms for the automated detection of security and privacy issues is required. Such an approach should be able to work in conditions of lack or inaccessibility of data and be able to evaluate the quality of the performed analysis. Moreover, such an approach should be modular and extensible, and it should take into account specific features of the smartphone-based driver monitoring systems.

In the following section, the data models that are used for the security and privacy analysis of the smartphone-based driver monitoring systems are presented in detail.

## 3. Data Models

Smartphone-based driver monitoring systems are complex structures that consist of multiple interconnected hardware and software elements that are working together on different tasks. As shown in Figure 1, those tasks can be aimed at the tracking of the driver behavior, interpretation of the inner and outer vehicle environment situation as well as integration with the vehicle infotainment systems for the data storage, transfer and/or extraction as well as connection to the remote services. Note that in the context of the smartphone-based system, all the hardware elements in Figure 1 are representing built-in sensors of the smartphone. In this work, the sensors of autonomous vehicles or external sensors that can be connected to the smartphone were not considered.

The approach presented in this paper is aimed at security and privacy analysis of driver monitoring systems that are based on smartphone sensors. Let us consider in more detail the main data models of the proposed approach.

### 3.1. Input Data

For the ease of understanding, it was decided to divide the data models of this section into input and output ones. The input data of the developed approach are divided into nine main objects and represented as follows:(1)in=(AM,FC,ag,AR,sc,lg,tf,ap,LR)
where

AM—set of attacker models, protection against which is considered during the analysis (for example, $am_{1}$ can be a simple attacker on mobile applications for casual use, $am_{2}$—advanced attacker on mobile applications for business use, $am_{3}$—powerful attacker on mobile applications for specific use, etc.);FC—functionality of the analyzed driver monitoring system (for example, $fc_{1}$ can be a road situation tracking, $fc_{2}$—context situation interpretation, $fc_{3}$—driver behavior tracking, etc.);ag—agreement between the user of the analyzed smartphone-based driver monitoring system and its owner that defines permissions on the user’s data extraction, storage and transferring;AR—access rights that are requested from the user of the system when the related application is installed on the user’s smartphone (usually, it is access to the smartphone’s camera ($ar_{1}$), microphone ($ar_{2}$) and storage ($ar_{3}$); note that such access rights might be given to the application even when it is not used, which defines additional security and privacy concerns [86]);sc—source code of the analyzed driver monitoring system;lg—logs of the analyzed driver monitoring system;tf—traffic of the analyzed driver monitoring system;ap—requirements of the mobile application that represents the analyzed smartphone-based driver monitoring system;LR—requirements under the law that define the work with the private data of the user of the analyzed driver monitoring system (usually depending on the country of the user).

The model of the attacker is necessary for the developed approach in terms of the balance between the security of the analyzed product and cost of the implementation of the protection measures. Obviously, security requirements for casual applications will be much lower than such requirements for applications that are used in critical infrastructure facilities. One of the possible ways to describe the possibility of the implementation of different classes of attacks is the introduction of the attacker model parameters, for example, such as types of access, knowledge and resources [87]. It means that any attacker $am_{i}\in AM$ can be represented as follows:(2)$$am_{i}=\left(ac^{am_{i}},kn^{am_{i}},rs^{am_{i}}\right),i\in 1..n,n\in N$$
where $ac^{am_{i}}$—access type of the attacker $am_{i}$ (for example, access through local or global network, physical access, etc.); $kn^{am_{i}}$—knowledge type of the attacker $am_{i}$ (for example, information from publicly available sources, knowledge about the parameters of the system or its hardware and software components, etc.); $rs^{am_{i}}$—resources type of the attacker $am_{i}$ (for example, widely spread tools and well-known vulnerabilities, specific tools and 0-day vulnerabilities, etc.).

In the developed approach, functionality defines the boundaries of the privacy and security issues detection process. To be able to do so, any functionality $fc_{i}\in FC$ can be represented in accordance with involved hardware components, software algorithms as well as data storage, extraction and transfer processes:(3)$$fc_{i}=\left(CN^{fc_{i}},AL^{fc_{i}},ST^{fc_{i}},EX^{fc_{i}},TR^{fc_{i}}\right),i\in 1..n,n\in N$$
where

$CN^{fc_{i}}$—hardware components of the smartphone that are involved in the implementation of the functionality $fc_{i}$ (for example, $cn_{1}^{fc_{i}}$ can be a rear camera, $cn_{2}^{fc_{i}}$—front-facing camera, $cn_{3}^{fc_{i}}$—gyroscope, etc.);$AL^{fc_{i}}$—software algorithms of the application that are involved in the implementation of the functionality $fc_{i}$ (for example, $al_{1}^{fc_{i}}$ can be a noise detection algorithm, $al_{2}^{fc_{i}}$—heavy rain detection algorithm, $al_{3}^{fc_{i}}$—driver drinking detection algorithm, etc.);$ST^{fc_{i}}$—data storage processes that are involved in the implementation of the functionality $fc_{i}$ (for example, $st_{1}^{fc_{i}}$ can be a smartphone local storage, $st_{2}^{fc_{i}}$—storage of the vehicle infotainment system, $st_{3}^{fc_{i}}$—cloud storage, etc.);$EX^{fc_{i}}$—data extraction processes that are involved in the implementation of the functionality $fc_{i}$ (for example, $ex_{1}^{fc_{i}}$ can be the extraction of the data from the application memory, $ex_{2}^{fc_{i}}$—user contacts, $ex_{3}^{fc_{i}}$—user media data, etc.);$TR^{fc_{i}}$—data transfer processes that are involved in the implementation of the functionality $fc_{i}$ (for example, $tr_{1}^{fc_{i}}$ can be USB, $tr_{2}^{fc_{i}}$—Bluetooth, $tr_{3}^{fc_{i}}$—cellular (4G, 5G, etc.), etc.).

Note that while each of the sets $CN^{fc_{i}}$, $AL^{fc_{i}}$, $ST^{fc_{i}}$ and $TR^{fc_{i}}$ can contain multiple elements, it is also possible that some of them would be empty. For example, some functionality does not require data storage or transfer processes, while another one does not involve hardware components into the implementation.

To continue our examples, let us consider the model description of one of the functionalities—a driver behavior tracking functionality ($fc_{3}$). Such a functionality involves the following elements:Hardware components ($CN^{fc_{3}}$): front-facing camera ($cn_{1}^{fc_{3}}$);Software algorithms ($AL^{fc_{3}}$): computer vision algorithms for the detection of the inappropriate behavior detection—drowsiness ($al_{1}^{fc_{3}}$), distraction ($al_{2}^{fc_{3}}$), unfastened belt ($al_{3}^{fc_{3}}$), eating ($al_{4}^{fc_{3}}$) and drinking ($al_{5}^{fc_{3}}$); note that the list of the algorithms for the driver behavior tracking depends on the product and may vary from one solution to another;Data storage processes ($ST^{fc_{3}}$): smartphone local storage ($st_{1}^{fc_{3}}$) and cloud storage ($st_{2}^{fc_{3}}$), assuming that machine learning models are too heavy to be used on a smartphone directly, so a remote server is required;Data transfer processes ($TR^{fc_{3}}$): cellular ($tr_{1}^{fc_{3}}$).

Summarizing, it means that the driver behavior tracking functionality can be represented as follows:$$fc_{3}=\left(\left\{cn_{1}^{fc_{3}}\right\},\left\{al_{1}^{fc_{3}},al_{2}^{fc_{3}},al_{3}^{fc_{3}},al_{4}^{fc_{3}},al_{5}^{fc_{3}}\right\},\left\{st_{1}^{fc_{3}},st_{2}^{fc_{3}}\right\},\left\{tr_{1}^{fc_{3}}\right\}\right)$$
and interpreted as the "driver behavior tracking involves smartphone front-facing camera data to detect driver drowsiness, distraction, unfastened belt, eating and drinking, while photos and videos are stored locally on the smartphone and transferred to the cloud data storage via cellular network”.

Agreement ag is usually presented as a text document which must be confirmed by the users of the product before the product becomes available. This document defines relations between the developers of the product and its potential users in terms of how the user’s personal data are extracted, stored and transferred, including situations when these data are provided to the third parties. In the developed model, such a document is represented as follows:(4)$$ag=\{sm_{1},...,sm_{n}\},n\in N$$
where $sm_{i}$—*i*-th statement in the agreement that was extracted from the document. Note that such statements may not include information related to the user’s personal data extraction/storage/transfer, and thus, the content of each statement must be additionally analyzed [88].

Another important part of the input data provided by the developers is the source code of their product. The source code of the mobile application is a complex structure that is represented as follows:(5)sc=(FL,PG,OB,VL,FU,IE)
where

FL—files that contain the source code of the application, including their extensions (for example, in Android Studio with Flutter project, $fl_{1}$ can be the *main.dart* file, $fl_{2}$—*project.yaml*, $fl_{3}$—*AppManifest.xml*, etc.);PG—packages that are used in the application (for example, $pg_{1}$ can be the *google_fonts* package, $pg_{2}$—*animations*, $pg_{3}$—*crypto*, etc.);OB—objects that are used in the source code (object-oriented programming);VL—variables that are used in the source code;FU—functions that are used in the source code;IE—IDEs (Integrated Development Environment) that were used during the development of the source code (for example, $ie_{1}$ can be the *Android Studio* IDE, $ie_{2}$—*DataGrip*, $ie_{3}$—*PyCharm*, etc.).

Any object $ob_{i}\in OB$ of the source code is represented as follows:(6)$$ob_{i}=\left(VL^{ob_{i}},FU^{ob_{i}}\right),i\in 1...n,n\in N$$
where $VL^{ob_{i}}$—set of variables of the *i*-th object of the application source code; $FU^{ob_{i}}$ set of functions of the *i*-th object of the application source code. For example, $vl_{1}^{ob_{i}}$ can be the *ANDROID_HOME* variable, $vl_{2}^{ob_{i}}$—*STUDIO_PROPERTIES* variable, $vl_{3}^{ob_{i}}$—*HTTP_PROXY* variable, etc. Meanwhile, $fu_{1}^{ob_{i}}$ can be the *databaseConnection* function, $fu_{2}^{ob_{i}}$—*userActionLogging* function, $fu_{3}^{ob_{i}}$—*widgetGeneration* function, etc.

Additionally, each variable besides the title must have its own data type and availability. It means that any variable $vl_{i}\in VL$ can be represented as follows:(7)$$vl_{i}=\left(tl^{vl_{i}},tp^{vl_{i}},ab^{vl_{i}}\right),i\in 1..n,n\in N$$
where $tl^{vl_{i}}$—title of the variable $vl_{i}$, it must be unique inside the variable’s data structure; $tp^{vl_{i}}$—data type of the variable $vl_{i}$ (INT, STR, TIMESTAMP, etc.); $ab^{vl_{i}}$—availability of the variable $vl_{i}$ (CONSTANT, PUBLIC, PRIVATE, etc.).

The same reasoning is applied to the representation of functions, but they can have their own variables in addition to the title, data type and availability. It means that any function $fu_{i}\in FU$ can be represented as follows:(8)$$fu_{i}=\left(tl^{fu_{i}},tp^{fu_{i}},ab^{fu_{i}},VL^{fu_{i}}\right),i\in 1..n,n\in N$$

Logs of the smartphone-based driver monitoring system are also an important source of the information about security and privacy issues. In terms of security, logs can be used for anomaly detection or event correlation [89], while in terms of privacy, logs allow checking which actions of the user are logged and how often this process occurs [90]. Moreover, it is important to check if user credentials are anonymized or used in logs directly. In the developed model, logs are represented as follows:(9)$$lg=\{ev_{1},...,ev_{n}\},n\in N$$
while each event $ev_{i}\in EV$ is represented as follows:(10)$$ev_{i}=\left(id^{ev_{i}},ts^{ev_{i}},sv^{ev_{i}},kw^{ev_{i}},dc^{ev_{i}}\right),i\in 1..n,n\in N$$
where $id^{ev_{i}}$—unique identifier of the event $ev_{i}$; $ts^{ev_{i}}$—timestamp of the event $ev_{i}$; $sv^{ev_{i}}$—severity of the event $ev_{i}$; $kw^{ev_{i}}$—keyword of the event $ev_{i}$; $dc^{ev_{i}}$—description of the event $ev_{i}$. As an example of the event structure, Syslog format can be used.

Together with logs, it is necessary to inspect the traffic of the smartphone-based driver monitoring system for security and privacy issues. In terms of security, traffic is used for attacks and anomalies detection. In terms of privacy, it is important to check what kind of the information about the user is presented in traffic, if is it anonymized and/or encrypted, how many recipients of such data can be detected, and so on. It means that in the developed model, traffic is represented as follows:(11)$$tf=\{pt_{1},...,pt_{n}\},n\in N$$
while each packet $pt_{i}\in PT$ can be represented as follows:(12)$$pt_{i}=\left(id^{pt_{i}},ts^{pt_{i}},sr^{pt_{i}},ds^{pt_{i}},pr^{pt_{i}},ln^{pt_{i}},pd^{pt_{i}}\right),i\in 1..n,n\in N$$
where $id^{pt_{i}}$—unique identifier of the packet $pt_{i}$; $ts^{pt_{i}}$—timestamp of the packet $pt_{i}$; $sr^{pt_{i}}$—source of the packet $pt_{i}$; $ds^{pt_{i}}$—destination of the packet $pt_{i}$; $pr^{pt_{i}}$—protocol of the packet $pt_{i}$; $ln^{pt_{i}}$—length of the packet $pt_{i}$; $pd^{pt_{i}}$—payload of the packet $pt_{i}$. As an example of the packet structure, the Wireshark tool representation can be used.

Another important type of information about the application is its hardware and software requirements for the correct work: necessary amount of the smartphone memory, supported models of smartphones and versions of their operating systems, supported models of vehicles and versions of their infotainment systems, etc. In the developed model, the application requirements are represented as follows:(13)ap=(ma,SN,VM,OS,IS)
where

ma—required amount of memory for the installation of the application;SN—set of smartphone models that are supported by the application (for example, $sn_{1}$ can be the iPhone 13 Pro smartphone, $sn_{2}$—Google Pixel 4a, $sn_{3}$—Xiaomi Redmi 10C, etc.);VM—set of vehicle models that are supported by the application (for example, $vm_{1}$ can be the Tesla Model Y vehicle, $vm_{2}$—Ford Mustang Mach-E, $vm_{3}$—Chevrolet Bolt EV, etc.);OS—set of operating systems with their versions that are supported by the application (for example, $os_{1}$ can be the Android 12 operating system, $os_{2}$—iOS 15.4.1, $os_{3}$—HarmonyOS 2.0.1.195 SP5, etc.);IS—set of infotainment systems with their versions that are supported by the application (for example, $is_{1}$ can be the Windows Embedded Automotive 7 infotainment system, $is_{2}$—Audi MMI 3G (Multi Media Interface), $is_{3}$—BMW iDrive 7, etc.).

Last but not the least, input data for the privacy issues detection is the information about the legislation in the field of working with private data in the country of the user of the smartphone-based driver monitoring system. For example, in the European Union, the data protection and privacy is regulated by the GDPR (General Data Protection Regulation [91]). Any requirement under the law $lr_{i}\in LR$ can be represented as follows:(14)$$lr_{i}=\left(oj^{lr_{i}},sj^{lr_{i}},md^{lr_{i}},st^{lr_{i}},ex^{lr_{i}},tr^{lr_{i}},pm^{lr_{i}}\right),i\in 1..n,n\in N$$
where

$oj^{lr_{i}}$—data object, work with which is covered by the requirement under the law $lr_{i}$ (for example, date of birth, name and surname, salary, etc.);$sj^{lr_{i}}$—data subject, whose work with the data object $oj^{lr_{i}}$ is covered by the requirement under the law $lr_{i}$ (for example, owner of the application, infotainment system, cloud, etc.);$md^{lr_{i}}$—method that is required to be used on the data object $oj^{lr_{i}}$ to be processed by the subject $sj^{lr_{i}}$ in accordance with the requirement under the law $lr_{i}$ (for example, encryption, anonymization, etc.);$st^{lr_{i}}$—data storage process of the data object $oj^{lr_{i}}$ that is allowed to the data subject $sj^{lr_{i}}$ in accordance with the requirement under the law $lr_{i}$;$ex^{lr_{i}}$—data extraction process of the data object $oj^{lr_{i}}$ that is allowed to the data subject $sj^{lr_{i}}$ in accordance with the requirement under the law $lr_{i}$;$tr^{lr_{i}}$—data transfer process of the data object $oj^{lr_{i}}$ that is allowed to the data subject $sj^{lr_{i}}$ in accordance with the requirement under the law $lr_{i}$;$pm^{lr_{i}}$—Boolean indicator that provides the information about the necessity of the permission from the application user for data subject $sj^{lr_{i}}$ to work ($md^{lr_{i}}$, $st^{lr_{i}}$, $ex^{lr_{i}}$, $tr^{lr_{i}}$) with data object $oj^{lr_{i}}$.

### 3.2. Output Data

The output data of the developed approach for security and privacy analysis are divided into six main objects and represented as follows:(15)ot=(SI,PI,SL,PL,MI,AQ)
where SI—security issues that were detected for the analyzed product; PI—privacy issues that were detected for the analyzed product; SL—security solutions that were suggested for the analyzed product; PL—privacy solutions that were suggested for the analyzed product; MI—information about the input data, most of which was detected by the developed approach; AQ—results of the quality evaluation of the product analysis process.

It is important to note that while the developers of the product are receiving only SL, PL and AQ as output of the security and privacy analysis approach, the provided information is formed in accordance with SI, PI and MI: SL is based on SI, PL→PI and AQ→MI.

Security issues (SI) can be detected in different parts of the source code of the smartphone-based driver monitoring systems as well as based on the model of the smartphone and version of its operating system, the model of the vehicle and version of its infotainment system, and so on. Such issues are describing classes of attacks to which the studied system is subject as well as detected vulnerabilities and weaknesses. It means that any security issue $si_{i}\in SI$ can be represented as follows:(16)$$si_{i}=\left(ij^{si_{i}},CA^{si_{i}},VN^{si_{i}},WN^{si_{i}}\right),i\in 1..n,n\in N$$
where $ij^{si_{i}}$—object of the $si_{i}$ (for example, version of the smartphone operating system, package of the Android application, function in the source code, etc.); $CA^{si_{i}}$—classes of attacks to which the $ij^{si_{i}}$ is susceptible (for example, broken access control, vulnerable and outdated components, etc.); $VN^{si_{i}}$—vulnerabilities to which the $ij^{si_{i}}$ is susceptible (for example, CVE-2022-28779, CVE-2022-0802, CVE-2021-39799, etc.); $WN^{si_{i}}$—weaknesses to which the $ij^{si_{i}}$ is susceptible (for example, CWE-200, CWE-297, CWE-921, etc.).

As well as security issues, privacy ones (PI) can be detected in various parts of the smartphone-based driver monitoring systems. More precisely, it is possible to detect violations of the privacy policy in accordance with the agreement (ag) between the user and the owner of the product, access rights (AR) requested by the mobile application as well as data privacy laws (LR) of the user’s country. Such violations can be found in the source code that is responsible for the data extraction, transfer and storage. It means that any privacy issue $pi_{i}\in PI$ can be represented as follows:(17)$$pi_{i}=\left(ij^{pi_{i}},ag^{pi_{i}},AR^{pi_{i}},LR^{pi_{i}}\right),i\in 1..n,n\in N$$
where $ij^{pi_{i}}$—object of the $pi_{i}$ (for example, function in the source code that is responsible for data extraction, storage, transfer, etc.); $ag^{pi_{i}}$—set of statements of the agreement ag, which were violated by the $ij^{pi_{i}}$; $AR^{pi_{i}}$—set of access rights requested by the application that are violating $ag^{pi_{i}}$ or $LR^{pi_{i}}$; $LR^{pi_{i}}$—set of requirements under the law, which were violated by the $ij^{pi_{i}}$.

It is important to note that while the detection of security and privacy issues is important, it is equally important to offer solutions to the issues found in order to provide system developers with additional support. In the developed model, any solution $sl_{i}\in SL$ to the detected security issues $SI^{sl_{i}}$ is represented as follows:(18)$$sl_{i}=\left(SI^{sl_{i}},SE^{sl_{i}}\right),i\in 1..n,n\in N$$
where $SI^{sl_{i}}$—security issues that can be covered by the $sl_{i}$; $SE^{sl_{i}}$—security measures that are required to cover $SI^{sl_{i}}$ (for example, data encryption/anonymization, use of escape symbols, access control, etc.).

Any solution $pl_{i}\in PL$ to the privacy issues $PI^{pl_{i}}$ is represented as follows:(19)$$pl_{i}=\left(PI^{pl_{i}},PE^{pl_{i}}\right),i\in 1..n,n\in N$$
where $PI^{pl_{i}}$—privacy issues that can be covered by the $pl_{i}$; $PE^{pl_{i}}$—privacy measures that are required to cover $PI^{pl_{i}}$ (data encryption/anonymization, access control, etc.).

Moreover, on each stage of the security and privacy issues analysis, it is necessary to check if all required data were provided by the developers. If some parts of the input data were missed, then the quality of the analysis will be reduced. In the developed model, information about missed input data is represented as follows:(20)$$MI=(mi^{FC},mi^{ag},mi^{AR},mi^{sc},mi^{lg},mi^{tf},mi^{ap},mi^{LR})$$
where

$mi^{FC}$—Boolean indicator that shows if the information about the functionality of the product was provided;$mi^{ag}$—Boolean indicator that shows if the information about the user agreement of the product was provided;$mi^{AR}$—Boolean indicator that shows if the information about the access rights on the user’s smartphone, which are required for the correct work of the product, was provided;$mi^{sc}$—Boolean indicator that shows if the information about the source code of the product was provided;$mi^{lg}$—Boolean indicator that shows if the information about the logs of the product was provided;$mi^{tf}$—Boolean indicator that shows if the information about the traffic of the product was provided;$mi^{ap}$—Boolean indicator that shows if the information about the system requirements of the product was provided;$mi^{LR}$—Boolean indicator that shows if the information about the requirements under the law to work with user’s data was provided.

Missed input data (MI) are used to evaluate the quality of the analysis that is represented as follows in the developed model:(21)$$AQ=(ql,\left(\left\{sg_{1},MI^{sg_{1}}\right\},...,\left\{sg_{n},MI^{sg_{n}}\right\}\right)),n\in N$$
where ql—quantitative metric that represents AQ in percentages (0% means no analysis, while 100% means full analysis); $sg_{i}$—*i*-th stage of the developed approach, i∈1..n; $MI^{sg_{i}}$—information about the input data that was missed during the *i*-th stage of the developed approach.

Thus, all data models that are used as the input and output for the developed approach were described in this section. In the following section, the main stages of the approach for the security and privacy analysis of the smartphone-based driver monitoring systems are presented in detail.

## 4. Proposed Approach

The developed approach for the security and privacy analysis of the smartphone-based driver monitoring systems from the developer’s point of view includes nine stages; see Figure 2. Let us consider each stage in more detail.

**Stage 0. The analysis of the functionality of the driver monitoring system to define the work process of the approach.** This stage is numbered as zero, because the functionality of the product (FU) determines which of the subsequent stages are involved in the analysis process as well as to what extent these stages can be implemented.

For example, during the early stages of the product development, there might be no text of the agreement (ag) between the owner of the product and its users as well as no information about the specific access rights on the user’s smartphone that are required for the correct functionality of the product (AR), which makes the analysis of the privacy issues ($PI_{3}$) at the fifth stage meaningless.

The third section of the paper describes that the data model of the product functionality (FU) consists of the smartphone hardware components (CN), application software algorithms (AL), data storage (ST), extraction (EX) and transfer (TR) processes. It means that based on the information about the functionality (FU) of the product, it is possible to assume which sensors (CN) of the smartphone are involved in the driver monitoring, what machine learning models are used for the intelligent image and video processing (AL) as well as what data are extracted from the smartphone (EX), where these data are stored (ST) and how they are transferred (TR).

The implementation of such connections implies the formalization of the description of the functionality of smartphone-based driver monitoring systems. This allows the approach to work with a fixed set of options, each of which can be logically linked to the appropriate components, algorithms and processes. At the same time, the set of components, algorithms and processes must also be formalized and limited. In addition, it is very important to use the abstract-detailed feature that was used in our previous work [92]. The main idea is to have a limited amount of abstract options for hardware components (for example, rear camera, GPS, gyroscope, etc.), software algorithms (for example, noise/driver drinking/heavy rain detection, etc.) and so on, while connecting those abstract options with their concrete implementations on the level of the detailed options (for example, an abstract rear camera, depending on the smartphone model, can be implemented as Apple iPhone 13 Pro Max/Google Pixel 6 Pro/Samsung Galaxy S22 Ultra rear camera, etc.).

Such a feature allows one to connect abstract options with general security issues, while detailed options can be connected with concrete vulnerabilities and weaknesses. It means that depending on the availability of the information about the concrete implementations, it is possible to provide a deeper analysis of the possible security issues.

Thus, the information about the functionality of the analyzed system is transformed into the involved hardware components (CN) and software algorithms (AL) as well as the data storage (ST), extraction (EX) and transfer (TR) processes; see Figure 3.

Hardware components (CN) and software algorithms (AL) are used as input for the first and second stages of the approach. During the first stage, they are used for detecting possible security issues, while the second stage is used to find out if security measures are already integrated into the product. A typical example of the hardware security element is the TPM (Trusted Platform Module) technology that is also used in mobile devices [93]. As an example of the software security element, any algorithm for encryption, authentication, access control, etc. is suitable.

**Stage 1. The detection of the security issues that are possible for the analyzed system.** Security issues ($SI_{1}$) are detected in accordance with the functionality (FU), access rights on the users smartphones provided to the product (AR), source code of the application (sc) as well as application’s requirements for the installation (ap); see Figure 4. During this stage, security issues are detected both for individual elements of the input data and based on their totality.

Firstly, security issues are detected in accordance with the functionality (FU) of the analyzed smartphone-based driver monitoring system. According to the developed model, the description of the functionality (FU) consists of the hardware components (CN), software algorithms (AL), data extraction (EX), storage (ST) and transfer (TR) processes. Meanwhile, the description of security issues ($SI_{1}$) consists of the issues object (ij) as well as related to this object classes of attacks (CA), vulnerabilities (VN) and weaknesses (WN).

It means that each $cn_{i}\in CN$, $al_{j}\in AL$, $ex_{k}\in EX$, $st_{l}\in ST$, $tr_{q}\in TR$ is represented as ij in $si_{p}^{1}\in SI_{1}$ and connected with $CA^{si_{p}^{1}}$, $VN^{si_{p}^{1}}$, $WN^{si_{p}^{1}}$. Such connections are possible based on the abstract-detailed feature and require a specific data or knowledge base to extract related security issues. Moreover, such a database must be kept updated and filled with all necessary data, or the quality of the analysis will decrease [94].

Thus, the following connections are analyzed during the first stage:CN,AL→CA,VN,WN, where the possibility of detection of VN and WN depends on the availability of the information about concrete implementations of the hardware components/software algorithms, while for the CA, abstract descriptions are enough;EX,TR,ST→CA, where any process that works with sensitive data is connected with the corresponding CA and defines an additional check of the product source code (sc) to identify specific VN and WN;AR→CA, where potentially dangerous user permissions to the application on the smartphone are connected with the corresponding CA;sc→VN,WN, where any part of the source code that works with the product user’s data is analyzed in terms of security to detect specific VN and WN;ap→VN,WN, where the availability of the information about a concrete model of the smartphone (SN) and vehicle (VM), as well as concrete version of the smartphone’s operating system (OS) and vehicle’s infotainment system (IS) provides a possibility to extract information about the corresponding VN and WN.

Note that the possibility of CA is detected in accordance with the inner classification of attacks that is using OWASP (Open Web Application Security Project) and extends it with physical level attacks on smartphone sensors; the possibility of VN is detected in accordance with the detected CVEs, while the possibility of WN is based on the detected CWEs.

It is known that the source code of the commercial product is a complex structure that may contain hundreds of thousands of code lines. That is why it is very important to identify specific operations in the source code and not analyze it entirely. To define the source code security analysis process in the approach, the functionality (FU) of the product is analyzed during the zero stage. The information about hardware components (CN) and software algorithms (AL), as well as data extraction (EX), storage (ST) and transfer (TR) processes helps the approach to form a step-by-step process of the source code (sc) analysis—what data are extracted from the smartphone (from sensors, from connected accounts, about the smartphone user, etc.), which algorithms are processing these data (machine learning, escaping, anonymization, etc.), and how it is stored (access control, encryption, hashing, etc.) and transferred (protocols, interfaces, networks, etc.).

In addition, to make the security issues analysis more dynamic, realistic and faster (do not protect from all known classes of attacks), it is required to use attacker models. Such models can define the possibility of the security issues through the parameters of the attacker, for example, the attacker’s types of knowledge, access and resources [95]. After that, it becomes possible to define through such parameters an attacker from which the analyzed system is required to be protected. At the same time, the model of the attacker can comply with the user’s country legal requirements, typical user’s requests, or requirements of the application’s catalog, which can help developers bring their product in line with these requirements.

**Stage 2. The detection of the security issues that are already covered by the analyzed system.** Covered security issues ($SI_{2}$) are detected in accordance with the functionality (FU) and the source code (sc) of the analyzed product; see Figure 5. This stage is mostly similar to the previous one, but covered security issues ($SI_{2}$) are detected only based on the source code (sc), while the functionality of the product (FU) as well as attacker models (AM) are defining the work process of such analysis.

During this stage, it is required in accordance with the security issues ($SI_{1}$) that were detected on stage 1 to find out which of them are already covered ($SI_{2}$) in the current implementation of the product (sc) with the help of security measures.

Availability of the information about $SI_{1}$ and $SI_{2}$ helps the approach to detect security issues ($SI=SI_{1}-SI_{1}⋂SI_{2}$) that are required to be covered in the analyzed smartphone-based driver monitoring system in accordance with the provided security requirements (based on AM).

It is important to note that in the implementation of the approach, the first and second stages can be developed as a single algorithm, but for the representation of the approach, it was decided to divide them into two individual stages.

**Stage 3. The analysis of the actual state of the user’s data extraction, storage and transfer processes in the product.** This stage aims to detect the privacy issues ($PI_{1}$) in accordance with the functionality (FU) of the driver monitoring system, namely, information about the data transfer (TR), extraction (EX) and storage (ST) processes, as well as the source code (sc), logs (lg) and traffic (tf) of the analyzed product; see Figure 6.

During this stage, the privacy issues are detected based on the answers to the following questions: “what private data are extracted?”, “how and where are private data stored?”, “how and where are private data transferred?”.

Similar to the first stage, during this one, it is required to identify specific operations that are working with the private data in the source code and do not analyze it entirely. And to define the source code privacy analysis process in the approach, the functionality (FU) of the product is analyzed during the zero stage.

Similar to the first stage, during this one it is required to identify specific operations that are working with the private data in the source code and do not analyze it entirely. To define the source code privacy analysis process in the approach, the functionality (FU) of the product is analyzed during the zero stage. Once again, the information about the data extraction (EX), storage (ST) and transfer (TR) processes helps the approach to form a step-by-step process of the source code (sc) analysis.

In addition, the product logs (lg) and traffic (tf) are used for the confirmation if some privacy issues were missed. For example, based on the sc analysis, it might be concluded that user’s private data are anonymized before storage and transfer, while based on the lg events, it might be found out that the user’s unique name or other credentials are presented as plain text.

It means that there might be errors during the source code analysis process, which requires both a manual check of the source code for the detection of the line(s) of code, which is the reason for privacy issues, and improvement of the automatic source code analysis module to not miss such privacy issues in the future.

**Stage 4. The analysis of the requirements that determine the work with private data in the user’s country.** During this stage, it is required to analyze text documents that are describing the product owner responsibilities and opportunities to work with a driver monitoring system user’s private data ($PI_{2}$); see Figure 7. It was decided to connect those requirements with the requirements of the user’s country, because in general, those requirements are already known and can be pre-analyzed and saved in the format necessary for the approach to work correctly.

Pre-analyzed requirements allow the approach to work faster. Moreover, the availability of the pre-analyzed privacy requirements from different countries or regions can help the developers tune their products for different markets separately. In addition, it is important to keep pre-analyzed requirements updated as well as store requirements of each version of the documents separately.

In situations when the country-based requirements are not known, the text document with such requirements can be uploaded by the developers for further analysis. Alternatively, requirements can be provided in the inner format of the approach to avoid precision and recall issues of the machine learning text analysis.

The process of the privacy issues analysis helps to transform sentences or paragraphs of the text into the data structure that was presented in the third section. Thus, it is required to detect parts of the text document that are defining what data objects (oj) can be processed by data subjects (sj) with or without the help of methods (md), with or without permission of the user (pm), as well as if it is allowed to extract (ex), store (st) and transfer (tr) those data objects. For example, the requirement under the law "The *name* of the *user* must be used in an *anonymized form* during *transferring* of the user experience data to *third parties*" can be represented as follows:$$lr_{1}=\left(name,thirdparty,anonymization,cloudstorage,null,cellular,0\right)$$
while for each additional data object such as the user’s surname, date of birth, nationality, etc., it is required to create a similar law requirement $lr_{2}$, $lr_{3}$, $lr_{4}$, etc.

During its work, the approach assumes that if there is no requirement that describes how to work with the user’s private data, then these data cannot be processed by the product owner or any other data subject. However, if the law requirements are fully unknown or not provided, then the approach assumes that everything is allowed and focuses only on the agreement between the user and the owner of the product.

**Stage 5. The analysis of user’s permissions to work with private data to the product owner and its partners.** During this stage, privacy issues ($PI_{3}$) are detected in accordance with the agreement between the user of the analyzed smartphone-based driver monitoring system and its owner (ag) as well as access rights that are requested from the user of the system when the related application is installed on the user’s smartphone (AR); see Figure 8.

The process of the agreement (ag) analysis is mostly the same as the process of the requirements under the law (LR) analysis during the previous stage. Once again, it is required to divide the text of the document into parts and find out those that are defining what data objects (oj) can be processed by data subjects (sj) with or without the help of methods (md), with or without permission of the user (pm), as well as if it is allowed to extract (ex), store (st) and transfer (tr) those data objects. The only difference is that the pm field is always true (with or without permission of the user) because of the nature of the analyzed document.

If the information about the agreement is provided by the developers not for the first time, it is also possible to work with pre-analyzed data instead of doing all the process once again. Although it might be suggested to do the analysis once again if since the last work with the approach, the machine learning model was improved, and thus, more correct results of the agreement analysis might be received (in case of the remote server, this analysis can be completed by the approach automatically). In addition, it is required to store information about each version of the agreement separately.

The access rights (AR) are analyzed in terms of their necessity for the correct work of the application, and if the over requesting is detected, then the corresponding privacy issues ($PI_{3}$) are added to the output of the stage. Note that even if the presence of the access rights request can be correct, the requested access right availability might be questionable (for example, access to the microphone is required for the noise detection that can distract the driver, but such access is required only during the use of the application, not for the whole time).

So, this stage provides information about the work with the user’s private data that was allowed by the user in accordance with the signed agreement. In addition, this stage detects if some access rights on the user’s smartphone are over requested during the installation of the analyzed application.

**Stage 6. The suggestion of the security measures to cover the detected security issues.** This stage aims to identify security measures that can cover the detected security issues (SI) and provide them to the developers as security solutions (SL); see Figure 9.

As was mentioned during the second stage description, the information about detected ($SI_{1}$) and already covered ($SI_{2}$) security issues helps the approach to identify security issues (SI) that are required to be covered in accordance with the provided security requirements. To do so, the approach forms a list of measures (SE) that can be used to cover SI. Then, a combination of the security issue(s) and measure(s) is provided to the developers as security solutions (SL).

It is important to note that connections between security issues and measures are required to be stored in the approach data or knowledge base. Such a storage should take into account that there might be a security issue that requires multiple measures, while one security measure might be able to cover multiple issues.

**Stage 7. The suggestions of the privacy measures to cover the detected privacy issues.** During this stage, it is required to analyze and combine privacy issues that were obtained during the third ($PI_{1}$), fourth ($PI_{2}$) and fifth ($PI_{3}$) stages of the approach as well as provide solutions (PL) to those issues based on the suggestion of different privacy measures; see Figure 10.

Thus, based on $PI_{1}$, the approach knows what happens with the private data of the user during the work of the analyzed application, while $PI_{2}$ defines what is possible regarding using the user’s private data in accordance with the law, and $PI_{3}$ defines if the agreement for certain operations was provided to the product owner. Additionally, $PI_{3}$ helps to detect access rights that are requested on the user’s smartphone during the installation of the application, but this is not necessary for its correct work.

So, to detect privacy issues that are required to be covered (PI), it is required:To compare $PI_{1}$ and $PI_{2}$ to identify which requirements under the law must be considered in detail by the approach ($PI_{12}$), because private data that are covered by them are processed in the application;To compare $PI_{12}$ and $PI_{3}$ to identify situations when private data are processed by the application without permission from the user, while such a permission is required according to the law (PI).

In addition, a comparison of $PI_{1}$ and $PI_{3}$ can be used to detect the incompatibility of the information from the agreement between the user and the product owner with the actual state of the work with the user’s private data.

Similar to the previous stage, after privacy issues (PI) are detected, the developed approach forms a list of measures (PE) that can be used to cover those issues. Such a combination of the privacy issue(s) and measure(s) is provided to the developers as privacy solutions (PL).

**Stage 8. The evaluation of the quality of the analysis based on the missed input data.** During this stage, it is required to evaluate the quality of the security and privacy analysis (AQ) in accordance with the input data that were not provided by the developers of the product (MI); see Figure 11. Because the input data can be missed only during the first six stages, the missed input data are divided into $MI_{0}$, $MI_{1}$, $MI_{2}$, $MI_{3}$, $MI_{4}$ and $MI_{5}$, correspondingly.

The situation with $MI_{0}$ is specific, because the only data that can be missed during zero stage are the information about the functionality of the analyzed product ($mi^{FC}$). To avoid the case, when the absence of this information means that there is nothing to analyze, it is required to use a representation of the functionality that is basic for most of the smartphone-based driver monitoring systems. Such a representation, obviously, reduces the depth of the approach’s security and privacy analysis, but it at least provides the information about typical privacy and security issues that are required to be taken into account during the development of such products.

During the first stage of the approach, it is possible to miss the information about the access rights on the user’s smartphone that are required for the correct work of the application (AR), the source code of the analyzed product (sc) and the installation requirements of the application for the user’s smartphone and vehicle (ap). It means that $MI_{1}$ indicates the absence or presence of these data through $mi^{AR}$, $mi^{sc}$, and $mi^{ap}$.

During the second stage of the approach, it is possible to miss the information only about the source code of the analyzed product (sc), which is indicated through $mi^{sc}$. It means that if the information about sc is missed, then the second stage of the approach is meaningless (output: none of the security issues are covered).

During the third stage of the approach, it is possible to miss the information about the source code (sc), logs (lg) and traffic (tf) of the analyzed driver monitoring system, which are indicated through $mi^{sc}$, $mi^{lg}$, and $mi_{tf}$. The most important data for this stage are sc, while lg and tf are used to clarify and confirm the information obtained on the basis of sc. However, in situations when sc is not available, lg and tf are becoming the only source of the information about the actual state of the user’s data extraction, storage and transfer processes in the product.

During the fourth stage of the approach, it is possible to miss the information only about the requirements under the law on work with user’s private data (LR), which is indicated through $mi^{LR}$. It means that if this information is not provided, then the analysis of this stage is meaningless, and it would not be possible to assess the legality of work with user’s private data in the analyzed product. However, it would still be possible to check if all manipulations with the user’s private data are mentioned in the agreement between the user and the owner of the analyzed product.

During the fifth stage of the approach, it is possible to miss the information about the agreement between the user and the owner of the product (ag) as well as the access rights on the user’s smartphone (AR), which are indicated through $mi^{ag}$ and $mi^{AR}$. As was mentioned during the fifth stage description, information about AR is supporting, but ag is more important for the analysis. Note that when ag and LR are not available, the privacy issues analysis becomes superficial and comes down to standard tips for anonymizing and encrypting sensitive data, while the presence of this information allows the approach to associate privacy measures with the need to comply with specific legal requirements or statements from the user agreement.

The evaluation of the analysis quality (AQ) shows in percentages how far the results are from the maximum possible value: 0% means no analysis, while 100% means full analysis; see Table 2.

Moreover, it is possible to separate the quality analysis for the security and privacy issues and output, for example, security—61.7%, privacy—50.0%, total—57.0% (situation, when sc and LR are not provided by the developers).

In addition to the quality numbers, it is equally important to provide to the developers the summary about the information that was missed during each stage of the approach and how it had affected the quality. It is assumed that such an output will help developers to take into account the risks associated with the inability to detect a number of security and privacy issues because of the lack of the input data.

Thus, all stages of the developed approach were described in this section. In the following section, an application of this approach to a passenger car driver monitoring use case is presented.

## 5. Experimental Evaluation

For the validation of the developed approach, it was decided to analyze the security and privacy of the smartphone-based driver-monitoring system that was intended to be used in a passenger car. Due to the early stage of the development of the analyzed system, the following input data were not provided: agreement between the user and the product owner (ag), legal requirements to work with user’s private data (LR) as well as logs (lg) and traffic (tf) of the analyzed product. In addition, the information about the source code (sc) was not provided due to its private nature.

Based on the missed input information, the approach evaluates the quality of the analysis as well as defines the self work process (some analysis stages could become meaningless because of lack of input data):Stage 1: $mi^{FC}=1$, $mi^{sc}=0$, $mi^{ap}=1$;Stage 2: $mi^{FC}=1$, $mi^{sc}=0$;Stage 3: $mi^{FC}=1$, $mi^{sc}=0$, $mi^{lg}=0$, $mi^{tf}=0$;Stage 4: $mi^{LR}=0$;Stage 5: $mi^{ag}=0$, $mi^{AR}=1$.

According to the missed input data, the approach concluded that the security and privacy analysis of the driver-monitoring system should be done based on stages 1, 2, 3 and 5, while stage 4 is meaningless.

To calculate the quality of the security and privacy analysis, the weighted values from Table 2 were used. Firstly, it is required to calculate the quality of the analysis per stage based on the stage value. This value indicates how important input data are for the analysis during the corresponding stage.

For example, during the first stage, the approach expects the information about functionality (FC), source code (sc) and installation requirements (ap) of the product as input data. The stage value of FC is 0.50, which means that the availability of the information about the functionality provides 50% of the analysis quality during this stage. According to those calculations, the following quality indicators were received for each stage:Stage 1: $mi^{FC}\times 0.50+mi^{sc}\times 0.30+mi^{ap}\times 0.20=0.35$, 35%;Stage 2: $mi^{FC}\times 0.20+mi^{sc}\times 0.80=0.20$, 20%;Stage 3: $mi^{FC}\times 0.10+mi^{sc}\times 0.50+mi^{lg}\times 0.20+mi^{tf}\times 0.20=0.10$, 10%;Stage 4: $mi^{LR}\times 1.00=0.00$, 0%;Stage 5: $mi^{ag}\times 0.70+mi^{AR}\times 0.30=0.30$, 30%.

The approach value from Table 2 defines how the quality of the stage analysis affects the total quality of security and privacy analysis of the developed approach. According to those values, stage 1 provides 50% of the quality, stage 2—10%; stage 3—20%; stage 4—10%; and stage 5—10%:$ql_{1}=0.350\times 0.500=0.175$, 17.5%;$ql_{2}=0.200\times 0.100=0.020$, 2%;$ql_{3}=0.100\times 0.200=0.020$, 2%;$ql_{4}=0.000\times 0.100=0.000$, 0%;$ql_{5}=0.300\times 0.100=0.030$, 3%.

After that, the quality of the analysis is calculated in total ($ql_{T}$) as well as separately for security ($ql_{S}$) and privacy ($ql_{P}$):$ql_{S}=\frac{ql_{1}+ql_{2}}{0.500+0.100}=\frac{0.175+0.020}{0.600=0.325}$, 32.5%;$ql_{P}=\frac{ql_{3}+ql_{4}+ql_{5}}{0.200+0.100+0.100}=\frac{0.020+0.000+0.030}{0.400}=0.125$, 12.5%;$ql_{T}=\sum_{i=1}^{5}ql_{i}=0.175+0.020+0.020+0.000+0.030=0.245$, 24.5%.

The results are showing that during early stages of the product development, it is difficult to provide security and privacy analysis of high quality, especially when the source code of the product as well as legal requirements to work with user’s private data and the agreement between the user and the product owner are missed.

According to the provided input data, the functionality of the analyzed driver monitoring system consists of the context situation interpretation ($fc_{2}$) and driver behavior tracking ($fc_{3}$). In accordance with those functionalities, the approach extracts the information about hardware components (CN), software algorithms (AL), data storage (ST), extraction (EX) and transfer (TR) processes that are necessary to provide $fc_{2}$ and $fc_{3}$. An example of such data extraction for $fc_{3}$ (driver behavior tracking) is presented in Section 3, while for the $fc_{2}$ (context situation interpretation), the results are as follows:Hardware components ($CN^{fc_{2}}$): microphone ($cn_{1}^{fc_{2}}$), accelerometer ($cn_{2}^{fc_{2}}$);Software algorithms ($AL^{fc_{2}}$): dangerous noise ($al_{1}^{fc_{2}}$), driver’s talking ($al_{2}^{fc_{2}}$) and dangerous maneuvers ($al_{3}^{fc_{2}}$) detection;Data storage processes ($ST^{fc_{2}}$): smartphone ($st_{1}^{fc_{2}}$) and cloud ($st_{2}^{fc_{2}}$);Data extraction processes ($EX^{fc_{2}}$): none;Data transfer processes ($TR^{fc_{2}}$): cellular ($tr_{1}^{fc_{2}}$).

Note that such components, algorithms and processes are extracted by the approach in accordance with inner representation of smartphone-based driver monitoring systems. That is why, after the extraction was made, the approach provides the developers a possibility to add or remove some abstract elements—not required hardware, software or data processes. In the analyzed use case, the following algorithms were removed from the product functionality, while hardware components and data processes were left unchanged:Context situation interpretation ($fc_{2}$): driver’s talking ($al_{2}^{fc_{2}}$);Driver behavior tracking ($fc_{3}$): driver’s eating ($al_{4}^{fc_{3}}$) and drinking ($al_{5}^{fc_{3}}$).

The final list of elements extracted based on the functionality is as follows:CN: Front-facing camera ($cn_{1}$), microphone ($cn_{2}$), accelerometer ($cn_{3}$);AL: Detection of the dangerous noise ($al_{1}$) and vehicle maneuvers ($al_{2}$), driver’s drowsiness ($al_{3}$), distraction ($al_{4}$) and unfastened belt ($al_{5}$);ST: Smartphone local ($st_{1}$) and cloud ($st_{2}$) storage;TR: Cellular ($tr_{1}$) network.

After the abstract elements are defined, the approach provides developers a possibility to add information about specific implementations of the hardware components and software algorithms, as well as data storage and transfer processes, that are used in their product. In the analyzed use case, the detailed information about hardware components can be extracted from the application’s installation requirements (ap), while the information about software algorithms is expected from the developers, because the source code is not available.

Input data showed that the following access rights are requested on the user’s smartphone when the analyzed product is installed: front-facing camera ($ar_{1}$); microphone ($ar_{2}$); accelerometer ($ar_{3}$); and local storage ($ar_{4}$).

Because of the lack of the source code of the product, it was not possible to check if those access rights are requested only during the work of the application or provided on an ongoing basis.

Hardware and software requirements for the correct work of the analyzed product were formulated as follows in the input data: 50 MB of the smartphone’s local memory (ma); Samsung Galaxy S7 (SN); and Android 8 (OS).

Sets of smartphone models (SN) as well as operating systems (OS) were provided with only one element because of the early stage of the product development—developers provided parameters of the device on which the prototype of the system is currently installed. Sets of vehicle models (VM) and infotainment systems (IS) were not fulfilled because the integration with in-vehicle systems is not developed in the current version of the product; thus, there are no requirements.

During the first stage of the approach, extracted CN, AL, ST and TR are analyzed to detect classes of attacks (CA), to which the analyzed system is susceptible. For the analyzed product, the possibility of the following CA was detected:Sensors (snr): generation of false events (gfe), bypass of the detection (bpd), physical harm (psh), replacement (rpl);Algorithm (alg): generation of false data (gfd), interception or modification of input data (imi), interception or modification of output data (imo), partial modification of the functionality (pmf);Application (app): insertion of the malicious code (imc), insertion of additional destinations for collected data (iad), insertion of malicious ads (ima);Operating system (ops): termination of security measures (tsm), spoofing of applications data (spf), extraction of user credentials (ruc), failure of update system (usf);External connections (exc): violation of the authentication system (vau), traffic sniffing (sff), man-in-the-middle (mim), interfaces jamming (jmm);External systems (exs): violation of the access control (vac), malfunction of API (mpi), cloud storage malfunction (clm), external sharing of data (smc).

To limit the amount of detected CA, it is required to analyze which of them are possible in accordance with that provided as the input model of the attacker. According to the provided model, intruders can attack the analyzed system only from the global networks (ac)—the driver is not considered as an intruder, they have knowledge about the analyzed system only from the publicly available sources (kn) while using only widely spread software tools and exploiting only known vulnerabilities (rs).

With such parameters of the attacker, the following CA are possible: ops={ruc,usf}; exc={vau,sff,mim,jmm}; and exs={vac,mpi,clm}.

Each possible class of attacks requires appropriate security measures, and the presence of such measures in the analyzed products is identified based on their source code during the second stage of the approach. Unfortunately, the source code was not provided due to its private nature, while the provided functionality of the product had no information about already applied security measures. It means that all classes of attacks detected during the first stage of the approach are assumed to be not covered in the current implementation of the product.

For further analysis of the security issues, it is required to analyze requirements for the correct work of the analyzed product (ap). According to those requirements, the analyzed product was connected with 653 vulnerabilities of Android 8 (cpe: 2.3: o: google: android: 8.0: *: *: *: *: *: * :*), see Table 3, as well as 211 vulnerabilities of Samsung devices; see Table 4.

Note that detected vulnerabilities (VN) must be further analyzed for their applicability to the current implementation of the product. If the important security updates are already installed, then the number of vulnerabilities will decrease significantly. Weaknesses (WN) in the analyzed product are detected in accordance with the detected vulnerabilities (VN), because in CVSS scores, CVEs are connected with CWEs. For example, CVE-2021-0316 (remote code execution over Bluetooth with no additional execution privileges needed in Android 8) is connected with CWE-787 (software writes data past the end, or before the beginning, of the intended buffer).

It means that the approach was able to detect the following security issues:Possibility of users’ credentials extraction, failure of the update system, violation of the authentication system, traffic sniffing, man-in-the-middle, interfaces jamming, violation of the access control, malfunction of API, cloud storage malfunction;653 CVEs for Android 8, 211 CVEs for Samsung devices and related CWEs, the relevance of which must be checked due to no information about already installed security updates.

To cover those issues, the following measures were suggested: installation of all available security updates, use of SSL (Secure Sockets Layer) and VPN (Virtual Private Network) tunnels, use of the latest cryptography libraries, and implementation of the high-level authentication. As for the privacy issues analysis, during the third and fifth stages of the developed approach, only two elements of the input data were provided:Functionality (FC) for the third stage; andRequired access rights on the user’s smartphone (AR) for the fifth stage.

Such a limited amount of the input data allows only to assume privacy issues, related to the data storage and transfer processes, that are known in accordance with FC. Additionally, AR are allowed one to assume what kind of sensitive data about the user can be collected based on the access to the smartphone’s microphone, front-facing camera and accelerometer.

Thus, the following privacy issues were detected:Presence of the user’s sensitive data in the smartphone local storage;Presence of the user’s sensitive data in the cloud storage;Transfer of the user’s sensitive data through the cellular network;Audio recording of the user;Photographing and video recording of the user;Recording of the user’s driving habits.

To ensure the privacy of user’s data, it is required to preprocess and anonymize user’s sensitive data before transmitting it to the cloud storage. Raw data records must be stored only on the smartphone of the user, if such a permission was provided to the application, and for a limited period of time. Moreover, data storage and communication channels must be secure.

## 6. Discussion

It is important to note that the developed approach is not aimed to replace experts in security and privacy analysis of mobile applications, cloud services, smartphone-based and driver monitoring systems, etc. We suppose that the presented solution would be useful for the development teams that are currently working without security and privacy experts and want to perform a preliminary analysis of their applications, so most of the issues can be fixed at early stages of their product lifecycle. In addition, this approach can also be useful for security and privacy experts due to the automatization of the routine tasks as well as offering security and privacy measures that might be different from those that are familiar to experts.

The advantages of the developed approach are as follows:Combination of the security and privacy analysis;Required protection level is set through the attacker model;Detected issues are reported together with appropriate measures;Approach can work in conditions of lack or inaccessibility of data;Specific features of driver monitoring systems are taken into account;Results of the analysis can be stored and used multiple times;Quality of the analysis is measured based on the missed input data;Approach is modular and extensible.

Meanwhile, the disadvantages of the developed approach are as follows:Correctness of the issues detection and measures suggestion highly depends on the completeness of the approach’s database;Fulfillment of such a database cannot be fully automated (manual work required);Quality of the security and privacy analysis directly depends on the input data that are provided by the developers;Security and privacy issues are only detected, not fixed.

The comparison of the developed approach with related works in accordance with data that are used as input and provided as output is presented in Table 5. It is assumed that such a comparison represents the difference in the functionality and areas of application of the approaches in the most indicative way.

Note that the representation of the input and output data is done based on the description of the data models from Section 3, while “+” and “–” are marked for each approach in general, without taking into account specific approaches that might use additional data as input or provide it as output.

In addition, note that the approach assumes that security and privacy issues in the products are not intentionally introduced but are the result of errors in the development process. It means that the main goal of the developed approach is to detect such situations automatically and provide useful feedback to developers about them.

## 7. Conclusions

In this article, two main scientific results are presented—the data models and the approach for security and privacy analysis of smartphone-based driver monitoring systems. Let us consider each result in more detail.

Unlike existing solutions, the developed data models are aimed at the representation of smartphone-based driver monitoring systems from the developer’s point of view for their subsequent security and privacy analysis. For the ease of understanding, it was decided to divide data models into input and output ones. The input data model includes nine main objects: set of attacker models, functionality of the analyzed system, agreement between the user of the system and its owner, application access rights on the user’s smartphone, source code, logs and traffic of the analyzed product, application installation requirements and legal requirements to work with user’s private data. The output data model includes six main objects: detected security and privacy issues, provided security and privacy solutions, information about missed input data and results of the quality evaluation of the product analysis.

The novelty of the approach lies in the combination of multiple analysis algorithms for the automated detection of the security and privacy issues and suggestion of the solutions to them. The developed approach consists of nine stages: analysis of the system’s functionality, detection of security issues that are possible and already covered in the analyzed system, analysis of the work processes with the user’s private data in the system, analysis of the legal requirements to work with user’s private data, analysis of the user’s permissions to work with private data, suggestion of the security and privacy measures as well as evaluation of the quality of the analysis based on the missed input data.

For the experiment, it was decided to validate the approach on a typical use case when the smartphone-based driver monitoring system is used in a passenger car. The analyzed system was at an early stage of the development; that is why some input data was missed, namely, the agreement between the user and the product owner, legal requirements to work with user’s private data as well as logs and traffic of the analyzed product. Moreover, the source code of the application was not provided due to its private nature. Such a situation reduced the quality of the analysis to the following values: security—32.5%, privacy—12.5%, total—24.5%. The approach was able to detect the following security issues: possibility of the extraction of the user credentials, failure of the update system, violation of the authentication system, traffic sniffing, man-in-the-middle, interfaces jamming, violation of the access control, malfunction of API, cloud storage malfunction, as well as multiple CVEs and CWEs related to them. The following privacy issues were detected: presence of the user’s sensitive data in the smartphone and cloud storage, transfer of this data via the cellular network, photographing, audio and video recording of the user, and recording of the user’s driving habits.

In future work, it is planned to improve and clarify the developed approach as well as perform the experimental evaluation of the approach on the system with a more detailed description. Moreover, it is planned to extend this approach with the security and privacy analysis from the user’s point of view, which uses black-box testing principles instead of white box ones. In addition, detailed experiments on security and privacy analysis of each part of the input data are planned as well. Finally, it is planned to consider other areas of application for the developed approach, because specific features of driver monitoring and smartphone-based systems are currently represented only through functionality and application requirements as well as the classification of possible security and privacy issues. We assume that it would be possible to implement the approach in such a way that will allow one to switch between different areas of application.

## Figures and Tables

**Figure 1 sensors-22-05063-f001:**
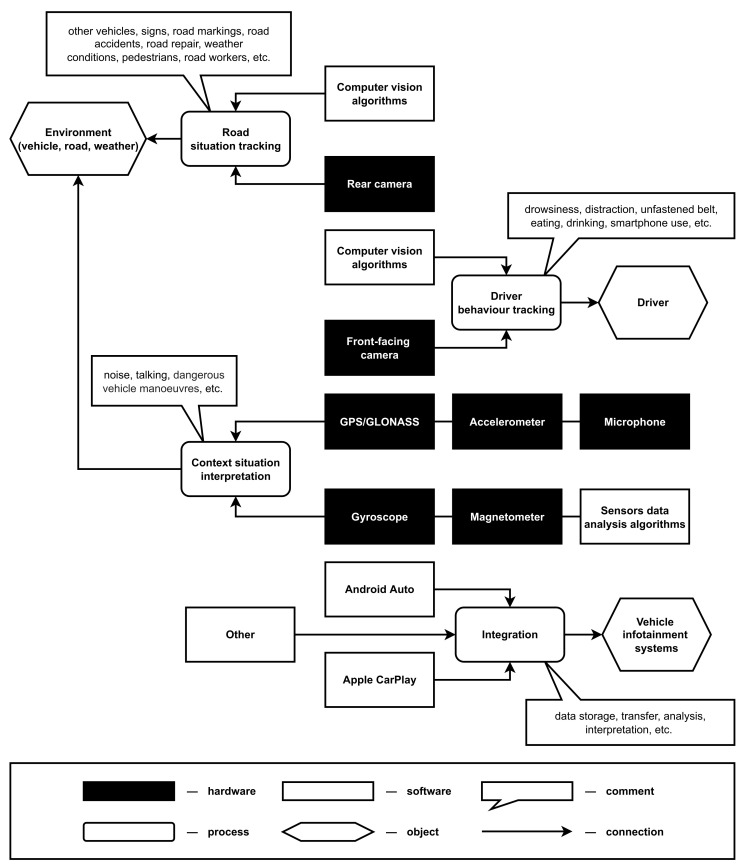
Structure of the smartphone-based driver monitoring systems.

**Figure 2 sensors-22-05063-f002:**
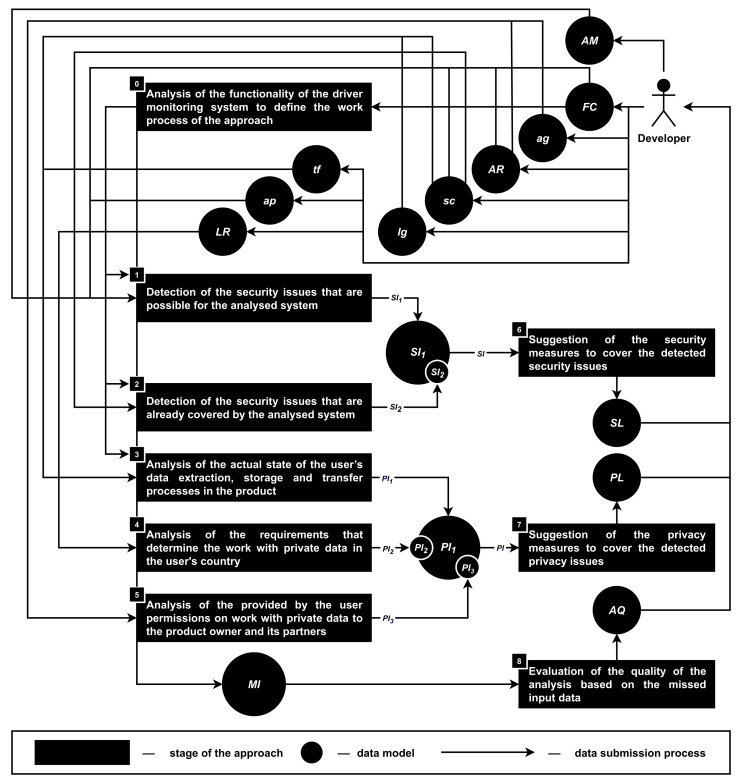
The approach for the security and privacy analysis of smartphone-based driver monitoring systems from the developer’s point of view.

**Figure 3 sensors-22-05063-f003:**
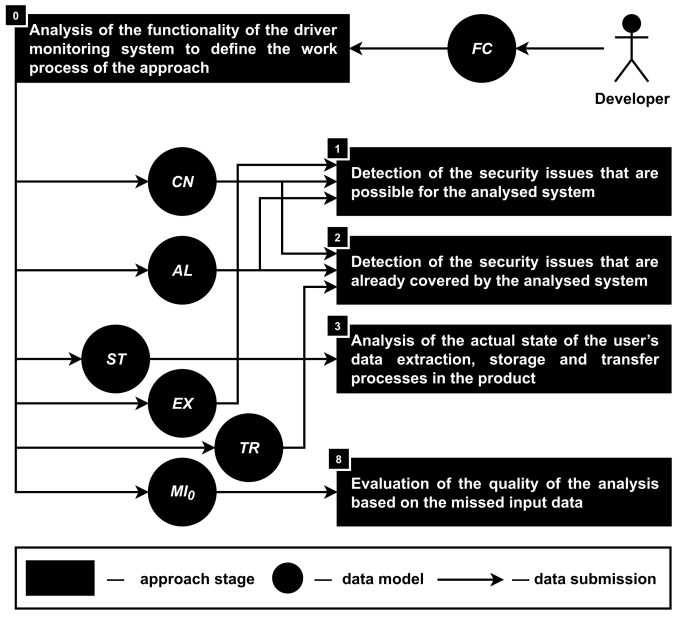
Stage 0—Analysis of the functionality of the driver monitoring system.

**Figure 4 sensors-22-05063-f004:**
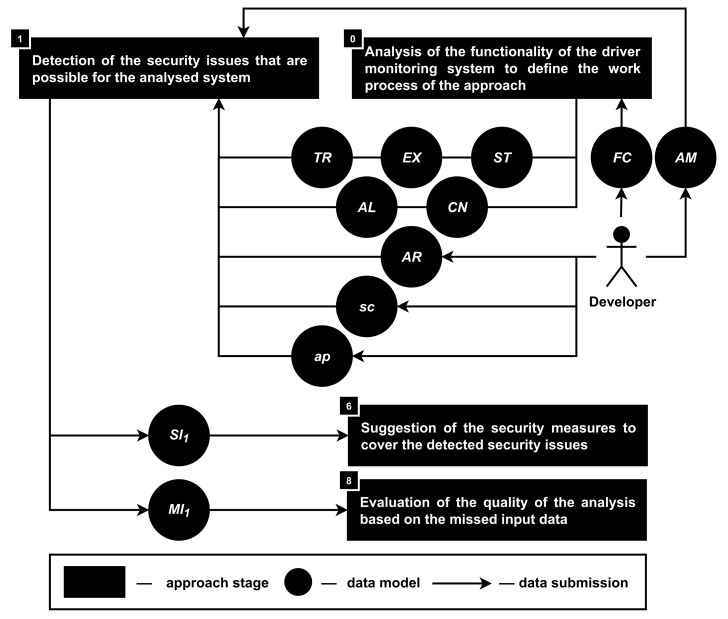
Stage 1—Detection of the possible security issues.

**Figure 5 sensors-22-05063-f005:**
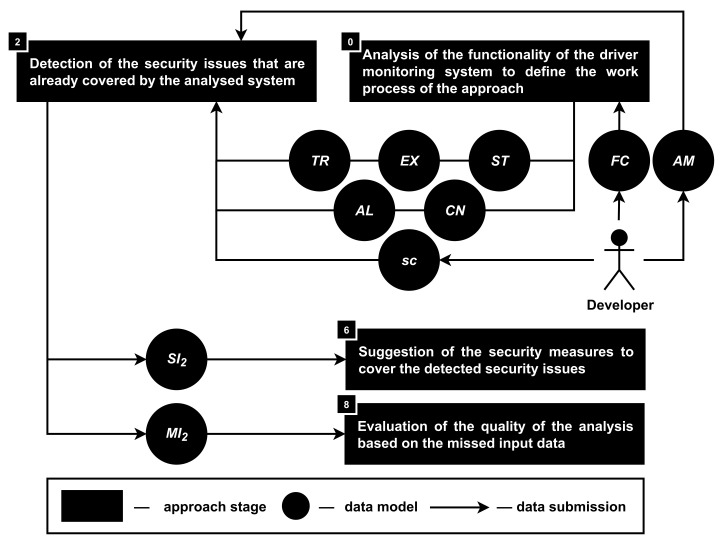
Stage 2—Detection of the covered security issues.

**Figure 6 sensors-22-05063-f006:**
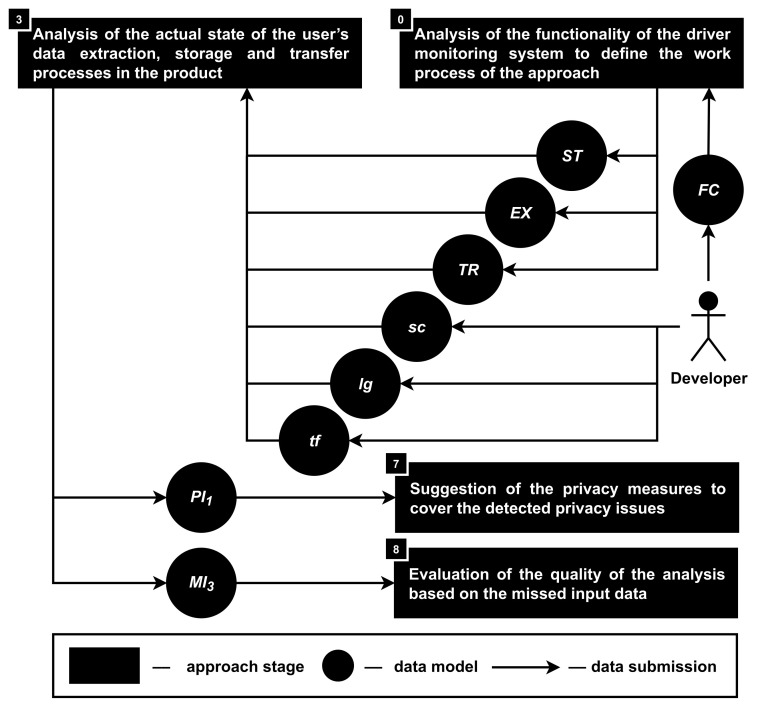
Stage 3—Analysis of the actual state of work with the user’s data.

**Figure 7 sensors-22-05063-f007:**
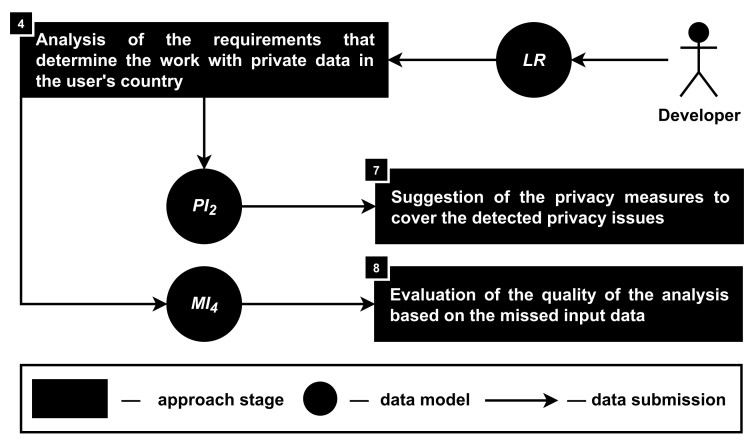
Stage 4—Analysis of the requirement to work with the user’s data.

**Figure 8 sensors-22-05063-f008:**
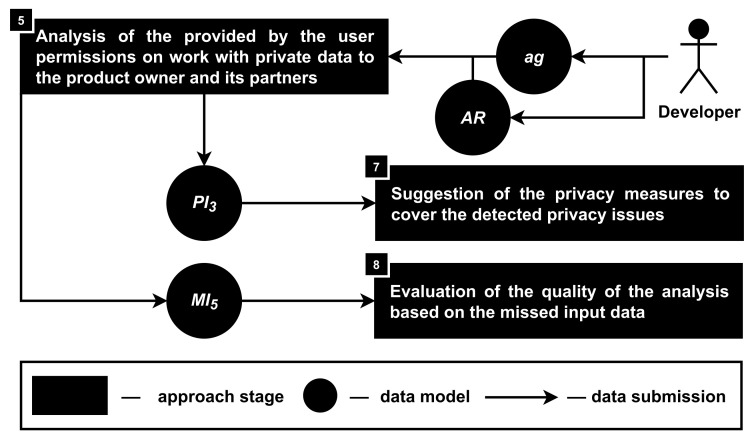
Stage 5—Analysis of the user’s permissions to work with private data.

**Figure 9 sensors-22-05063-f009:**
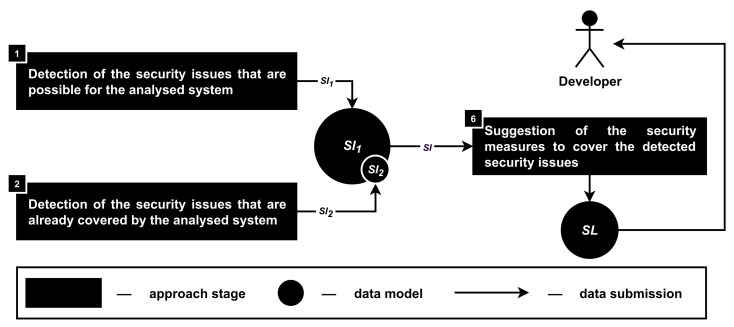
Stage 6—Suggestion of the security measures.

**Figure 10 sensors-22-05063-f010:**
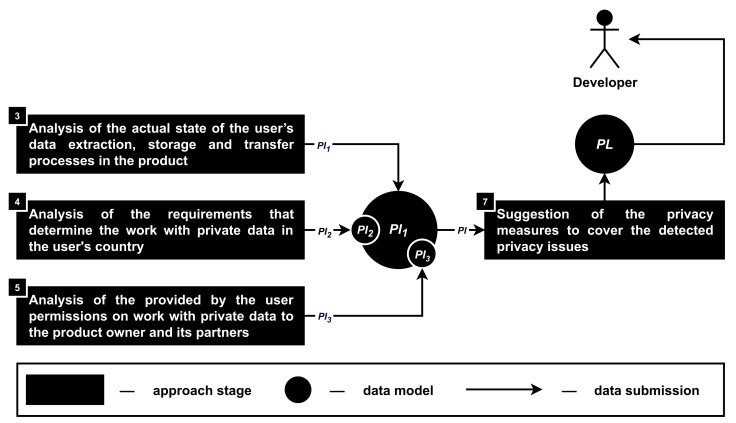
Stage 7—Suggestion of the privacy measures.

**Figure 11 sensors-22-05063-f011:**
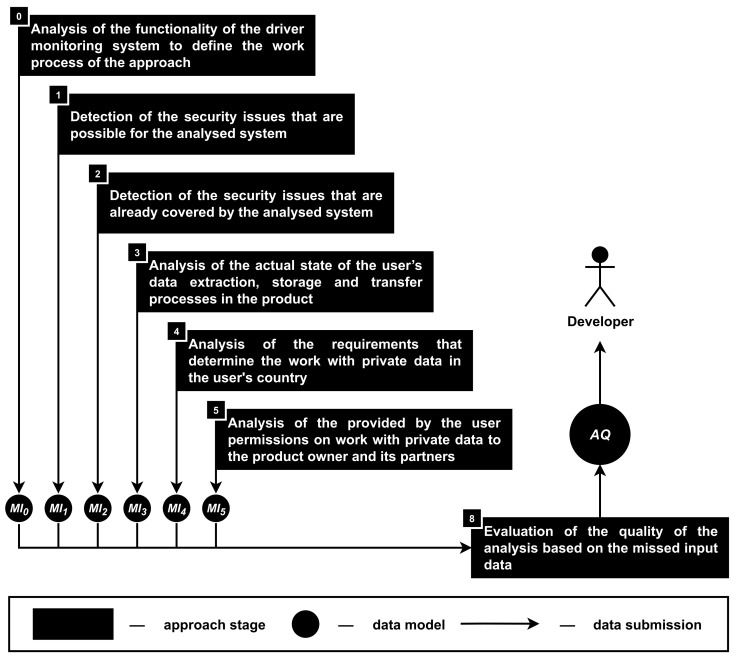
Stage 8—Evaluation of the analysis quality.

**Table 1 sensors-22-05063-t001:** Approaches’ contribution.

Object of Analysis	References	Analyzed Data	Provided Data
Functionality	[31,32,33,34,35,36,37,38,39,40,41,42,43,44,45,46]	hardware and software elements, interfaces, data transfer protocols, data extraction, storage and transfer processes	security threats, classes of attacks
Configuration	[18,47,48,49,50,51,52,53]	platforms and versions of their hardware components, firmware and operating systems, as well as software applications used by them	vulnerabilities, weaknesses, risks
Source code	[59,60,61,62,63,64,65,66]	code’s architecture and logic of operation, extraction, storage or transfer of data processes, interactions between elements	detected buffer overflows, memory leaks and code inserts, compliance with privacy policy
Logs	[69,70,71,72,73]	events	anomalies, attacks, leakage of user’s sensitive data
Traffic		packets	
Documents	[74,75,76]	agreements, legal documents	current state of work with user’s private data and its compliance with legal requirements

**Table 2 sensors-22-05063-t002:** Weighted values of the input data importance.

Issue	Stage	Input	Stage Value	Approach Value
SI	1	$mi^{FC}$	0.50	0.50
		$mi^{sc}$	0.30	
		$mi^{ap}$	0.20	
	2	$mi^{FC}$	0.20	0.10
		$mi^{sc}$	0.80	
PI	3	$mi^{FC}$	0.10	0.20
		$mi^{sc}$	0.50	
		$mi^{lg}$	0.20	
		$mi^{tf}$	0.20	
	4	$mi^{LR}$	1.00	0.10
	5	$mi^{ag}$	0.70	0.10
		$mi^{AR}$	0.30	

**Table 3 sensors-22-05063-t003:** Android 8 vulnerabilities.

	2017	2018	2019	2020	2021	Total
Denial of service	9	21	2	7	1	40
Code execution	20	43	38	43	3	147
Overflow	14	12	9	32	2	69
Memory corruption	0	1	11	4	1	17
SQL injection	0	1	3	5	0	9
Cross site scripting	0	1	0	0	0	1
Directory traversal	0	2	0	3	0	5
Bypass	0	11	12	35	4	62
Gain information	21	27	7	32	0	87
Gain privileges	0	0	1	3	1	5

**Table 4 sensors-22-05063-t004:** Samsung’s vulnerabilities.

	2017	2018	2019	2020	2021	Total
Denial of service	5	0	0	1	3	9
Code execution	3	13	1	1	6	24
Overflow	1	4	1	1	7	14
Memory corruption	0	1	0	0	0	1
Cross site scripting	1	3	4	0	1	9
Directory traversal	2	1	0	0	1	4
Bypass	2	0	0	0	2	4
Gain information	11	2	0	0	4	17
Gain privileges	2	0	0	1	0	3
Cross site request forgery	0	1	0	0	0	1

**Table 5 sensors-22-05063-t005:** Approaches’ comparison.

Approach	Input Data									Output Data					
	AM	FC	ag	AR	sc	lg	tf	ap	LR	SI	PI	SL	PL	MI	AQ
Functionality [31,32,33,34,35,36,37,38,39,40,41,42,43,44,45,46]	+	+	–	–	–	–	–	–	–	+	–	+	–	–	–
Configuration [47,48,49,50,51,52,53]	–	+	–	–	+	–	–	+	–	+	–	+	–	–	–
Source code [59,60,61,62,63,64,65,66]	–	–	–	+	+	–	–	–	–	+	+	+	+	–	–
Logs, traffic [69,70,71,72,73]	–	–	–	–	–	+	–	–	–	+	+	–	–	–	+
	–	–	–	–	–	–	+	–	–	+	+	–	–	–	+
Documents [74,75,76]	–	–	+	+	–	–	–	–	+	–	+	–	+	–	+
Developed	+	+	+	+	+	+	+	+	+	+	+	+	+	+	+

## Data Availability

Not applicable.

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
