# Peer review of "Security and Privacy Analysis of Smartphone-Based Driver Monitoring Systems from the Developer’s Point of View"

_sensors, 2022, doi:10.3390/s22135063_

Round 1

Reviewer 1 Report

In this paper the authors propose a security an privacy analysis framework for smartphone-based driver monitoring systems. The proposed approach performs analysis from the developer's point of view.

The paper presents a good approach with sufficient testing. Only a few minor issues need to be addressed before the paper becomes ready for publications.
The data modelling and the proposed approach are well written and easy to follow, although lengthy.

Minor Comments:
1. The abstract is expected to give some context of  the problem before diving into the proposed system.
2. It would be helpful to create a literature summary table showing the different features and contributions of related works in Section 2.
3. It is necessary to provide a detailed comparison of the proposed work performance and features with the related works. A table with proper analysis is required in Section 6 to establish the superiority of the proposed system over other works.
4. The paper could use some proofreading to improve its readability.

Author Response

Dear reviewer, please, find our response to your review in the attached document.

Reviewer 2 Report

Overall:

The manuscript described an automated approach to conducting security and privacy analysis of driver monitoring systems based on smartphone sensors for product improvements, aiming to help developers improve their driver monitoring systems. The approach is based on a white-box principle and is designed to be modular and extensible, which seems to follow software engineering practice. The authors close with suggestions that detected potential security and privacy issues and discussed their mitigation, together with limitations of the analysis due to the scarcity of data. The study presented in this manuscript is very interesting and important under the current trend of mobile VGI (crowdsourcing), CAVs, and edge computing in developing intelligent transportation systems and smart city apps. The manuscript is well-organized, well-written, and easy to read. The methodology is reasonable and supported by a cause study. I feel like the current manuscript can be further shortened to be more concise. I am in support of this manuscript, and I only have a few comments.

Comment 1:

In lines 35-36, the author mentioned, “Moreover, use of smartphones as well as cloud services introduces new security and privacy risks to the system [4,5]”. I think it is important to explain what data and information are considered users’ privacy. Location data from GPS coordinates without user identity are not considered privacy in many studies, and many companies, like Wejo and safe graph, do provide services to collect vehicle trajectories. The authors should make a list and discuss what mobile-sensed data are considered private or sensitive.

Comment 2:

From lines 52-54, the authors mentioned, “Therefore, this work is aimed at developing the original approach for security and privacy analysis of smartphone-based driver monitoring systems from the developer’s point of view”. I wonder if the analytical approach for different mobile app development platforms is universal or different? Do Android and IOS face similar app security and data privacy issues? Does existing mobile app SDK (e.g., react-native, angular-native) define conventions and practices to resolve these security issues? 

Comment 3:

In lines 96-98, the authors mentioned different analyses that focus on different components, including hardware, software elements, interfaces, data transfer protocols, data extraction, storage, and transfer processes. I think these components are associated with the smartphone app itself, which is often considered the client-side of the application. However, the server-side or cloud-side application that provides web-based APIs, sockets, and online datastore to support a smartphone app can also be vulnerable to data leaks and attacks. Do you think it is necessary or relevant to discuss potential security risks that are associated with the backend applications?

Comment 4:

Line 183-186 talk about how different vehicle components are connected into a workflow to track and monitor the driver’s behavior. The authors also refer to Figure 1 to show the workflow. I am confused with some of the components displayed in this Figure. I understand that a smartphone has GPS, accelerometer, and gyroscope, while the authors also put other sensors (e.g., rear camera, front-facing camera, and computer vision algorithms) that seem to belong to the vehicle into the Figure. Are we talking about the phone app? Or connected autonomous vehicles? Are these built-in sensors of a smartphone? Or external sensors that are connected to a smartphone app through Bluetooth or USB?

Comment 5:

Section 3 provides very comprehensive explanations and reviews of smartphone data models, which is very good. I feel like the author could reasonably shorten some of the discussion by only presenting content that is relevant to the design requirement of the proposed analytical approach.

Author Response

(The authors gave the same response as above.)
